# The Combined Therapy of Cabozantinib, Crizotinib, and Osimertinib in a Lung Cancer Patient with Acquired MET Amplification and Resistance Mutations

**Balázs Jóri** [1,2,*], **Christine Vössing** [1,2], **Judith Pirngruber** [1,2], **Eva Maria Willing** [1,2], **Kathrin Arndt** [1,2], **Markus Falk** [1,2], **Markus Tiemann** [2], **Lukas C. Heukamp** [1,2] **and Petra Hoffknecht** [1,3]

1   Lungenkrebsmedizin Oldenburg, Georgstraße 12, 26121 Oldenburg, Germany; arndt@hp-hamburg.de (K.A.); heukamp@hp-hamburg.de (L.C.H.)
2   Institut für Hämatopathologie Hamburg, Fangdieckstraße 75A, 22547 Hamburg, Germany
3   Department of Thorax Oncology, Niels-Stensen-Kliniken, Franziskus-Hospital Harderberg Alte, Rothen-Felder Straße 23, 49124 Georgsmarienhütte, Germany
*   Correspondence: jori@hp-hamburg.de

**Abstract:** EGFR-mutant lung cancers develop a wide range of potential resistance alterations under therapy with the third-generation EGFR tyrosine kinase inhibitor osimertinib. MET amplification ranks among the most common acquired resistance alterations and is currently being investigated as a therapeutic target in several studies. Nevertheless, targeted therapy of MET might similarly result in acquired resistance by point mutations in MET, which further expands therapeutic and diagnostic challenges. Here, we report a 50-year-old male patient with EGFR-mutant lung adenocarcinoma and stepwise acquired resistance by a focal amplification of MET followed by D1246N (D1228N), D1246H (D1228H), and L1213V (L1195V) point mutations in MET, all detected by NGS. The patient successfully responded to the combined and sequential treatment of osimertinib, osimertinib/crizotinib, and third-line osimertinib/cabozantinib. This case highlights the importance of well-designed, sequential molecular diagnostic analyses and the personalized treatment of patients with acquired resistance.

**Keywords:** osimertinib; crizotinib; cabozantinib; sequential therapy; resistance; c-MET; focal amplification; copy number calling; D1246N; D1228N; D1246H; D1228H; L1213V; L1195V; hybrid capture; NGS

## 1. Introduction

Osimertinib, an irreversible epidermal growth factor receptor (EGFR) tyrosine kinase inhibitor (TKI) of the third generation, has been approved as a first line therapy for EGFR-mutated non-small-cell lung cancer (NSCLC) [1]. However, under osimertinib treatment pressure, tumors activate alternate signaling pathways and develop a broad range of resistance [2]. Among those, the amplification of the mesenchymal–epidermal transition (MET) gene has been reported to occur in 7–15% of patients treated with osimertinib as first-line treatment [3].

MET activates the ERBB3/PIK3/AKT signaling pathway, and targeting MET may therefore open up second-line therapeutic opportunities [4]. The multi-target ALK/MET/ROS1 inhibitor crizotinib induced substantial tumor shrinkage confined to patients with MET amplifications according to the PROFILE 1001 study [5], and the effect of combination treatment has also been evaluated in retrospective studies [6]. Responses have also been reported in patients with the combined treatment of osimertinib and the MET-TKI capmatinib [7,8] or erlotinib and crizotinib [9].

The use of crizotinib also results in resistance after several months, and among the resistance mechanisms, point mutations in the MET gene, such as p.D1246N (also known as p.D1228N), emerge [10–13]. According to case reports, in these patients, type II MET-TKI cabozantinib might overcome crizotinib resistance [14,15]. However, as of today,

limited data are available on sequentially treated patients with stepwise acquired resistance mechanisms and treatments.

Here, we report a male EGFR-positive NSCLC patient with a profound clinical response to the combined and sequential therapy of osimertinib, then due to an acquired focal MET amplification by combined osimertinib/crizotinib, followed by a combination of osimertinib/cabozantinib due to an acquired MET D1246N resistant mutation. These treatment combinations led to the acquisition of the additional MET point mutations D1246H and L1213V, as assessed only by hybrid-capture next-generation sequencing (HC-NGS). The therapy resulted in an overall survival of 31 months.

## 2. Case Presentation

We describe a 50-year-old, never-smoking, male NSCLC patient of Asian origin. The patient's history is summarized in Figure 1, while the molecular results are displayed in Table 1.

**Table 1.** Applied molecular assays and detected alterations. Detected molecular alterations with NGS. Of note, the nomenclature of MET point mutations depends on the applied transcript number.

| Date | | | | April 2020 | January 2021 | December 2021 | January 2022 | May 2022 | May 2022 |
|---|---|---|---|---|---|---|---|---|---|
| Assay | | | | NEO-Plus | NEO-Select | NEO-Liquid | HS2-Lung | HS2-Lung | HS-Liquid |
| Sample Type | | | | FFPE | FFPE | Eff. | FFPE | FFPE | Cf |
| **Point Mutations detected with HC-NGS** | | | | | | | | | |
| **Gen** | **Transcript.** | **DNA-change** | **Protein change** | **Variant Allelic Frequency** | | | | | |
| EGFR | NM_005228.4 | c.2236_2250del | p.E746_A750del | 17% | 56% | 7% | 27% | 39% | 60% |
| TP53 | NM_001126112.2 | c.559 + 1G > A | p.? | 12% | 61% | 7% | 15% | 31% | 41% |
| MET | NM_00112750.3 (NM_000245.4) | c.3736G > A (c.3682G > A) | p.D1246N (p.D1228N) | - | - | 0.1% | - | - | 0.7% |
| MET | NM_00112750.3 (NM_000245.4) | c.3736G > C (c.3682G > C) | p.D1246H (p.D1228H) | - | - | - | - | - | 0.4% |
| MET | NM_00112750.3 (NM_000245.4) | c.3637C > G (c.3583C > G) | p.L1213V (p.L1195V) | - | - | - | - | - | 0.5% |
| **Amplifications detected with HC-NGS** | | | | | | | | | |
| **Gen** | **Transcript** | | | **Level [1]** | | | | | |
| EGFR | NM_005228.4 | | | - | + | - | + | + | n.a. |
| MET | NM_00112750.3 | | | - | ++ | ++ | ++ | ++ | n.a. |
| **Additional alterations detected with HC-NGS** | | | | | | | | | |
| Tumor Mutational Burden (Muts/mB) | | | | 17.42 | - | - | - | - | - |
| Microsatellite Instability | | | | MSI-L | - | - | - | - | - |
| TP53, NRAS, BRCA2, and ARAF | | | | - | DEL | - | - | - | - |

[1] +: low-level amplified, ++: focally amplified, -: not detected, n.a.: not included in the assay, DEL: gen deletion, MSI-L: microsatellite-low, FFPE: formalin-fixed, paraffin-embedded tissue material, Eff.: pleural effusion, Cf.: cf-DNA.

He was diagnosed with stage IVB lung adenocarcinoma with initial pathological staging cT4, pN3, cM1c, and UICC in April 2020 with metastases in the liver, bone, and both retinae. In the formalin-fixed, paraffin-embedded (FFPE) tissue sample from the primary tumor, HC-NGS identified a typical exon 19 deletion in EGFR and a likely inactivating TP53 mutation with a potential splice effect at the donor site of exon 5 [16]. There were no copy number changes, but low degree microsatellite instability (MSI-L) (one out of five microsatellite loci) and elevated tumor mutational burden (TMB) were also detected (17.42 Muts/MB) [17,18]. Immunohistochemistry (IHC) showed weak PD-L1 expression (TPS: 5%, positive immune cells: 2%, CPS: 7%). The patient started on osimertinib orally at a daily dosage of 80 mg. In May 2020, partial pulmonary remission was observed, and later in August 2020, hepatic metastases and pulmonary metastases were regredient as well.

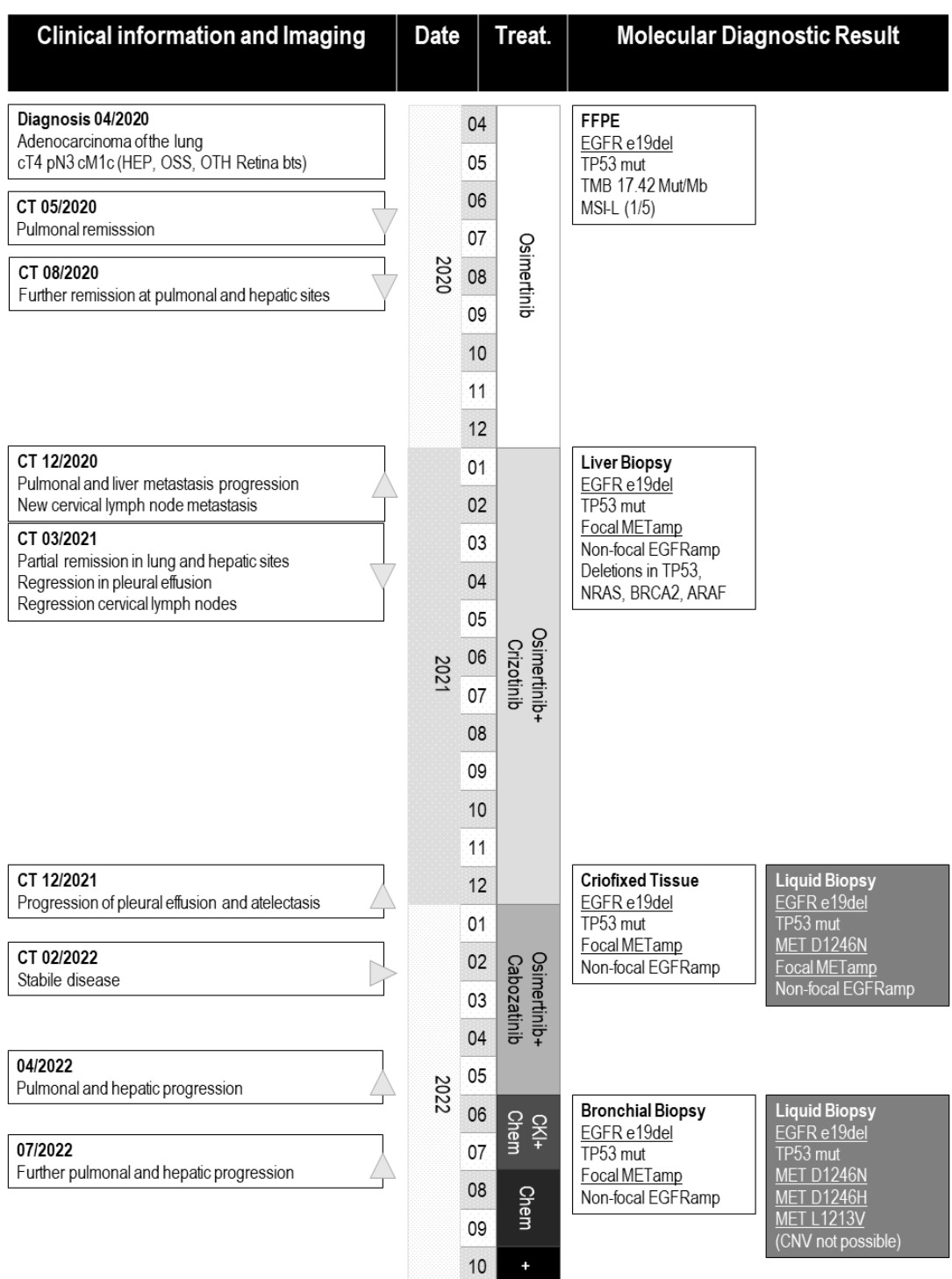

**Figure 1. Patient history**. ▲: progression, ▼: regression, ►: stabile disease, CKI: checkpoint inhibition, chem: chemotherapy, OSS: osseous metastasis, HEP: hepatic metastasis, OTH: ophthalmic metastasis, treat.: treatment, +: deceased. Biomarkers that were used for stratification are underlined (i.e., "EGFR e19del").

In January 2021, after 9 months of initial first-line treatment with osimertinib, a second molecular analysis was performed from the progredient hepatic metastases, which confirmed the original EGFR and TP53 mutations and revealed additional deletions in TP53, NRAS, BRCA2, ARAF, a low-level EGFR, and a focal MET amplification. As MET-amplified NSCLC has been reported as sensitive to the small-molecule TKI crizotinib in multiple

studies [5,19], the patient started to receive the combined treatment of osimertinib (80 mg) with crizotinib (250 mg). Six weeks later (March 2021), the patient showed partial remission in both lung and hepatic sites and regression in pleural effusion and cervical lymph nodes (Figure 2). As side effects, slight cutaneous toxicity was observed; besides this, the patient showed a good appetite and stable weight as part of a routine health status assessment in July 2021.

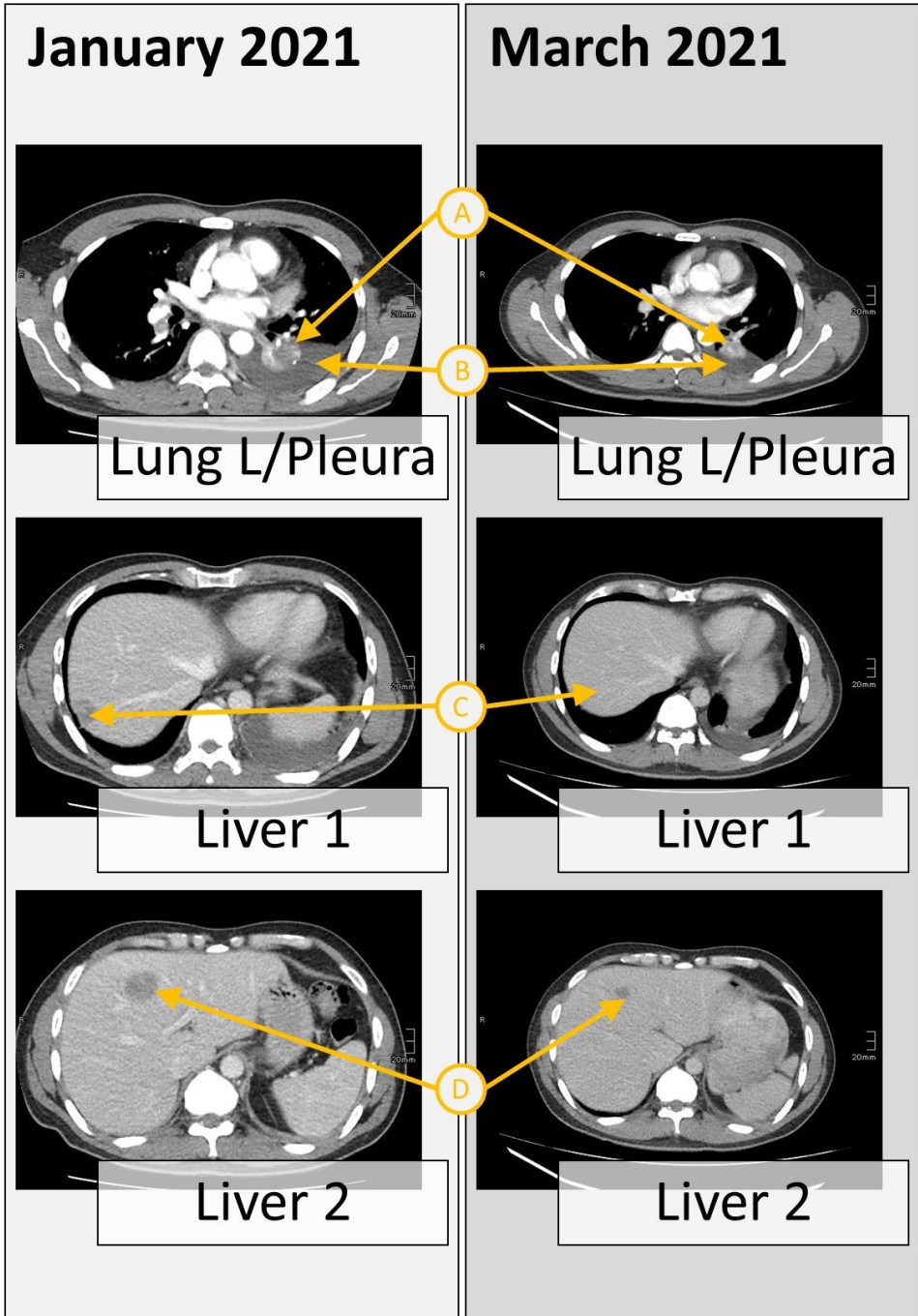

**Figure 2.** Response to the combined treatment of osimertinib with crizotinib of multiple sections. Computer tomography scan from the chest and liver sections at the time of tumor progression after osimertinib, revealing multiple metastases (January 2021), and after 6 weeks of therapy with crizotinib plus osimertinib, revealing partial remission in both lung and hepatic sites. The reduction can be seen in tumor size (A), pleural effusions (B), and the liver metastatic sites (C and D).

In early December 2021, the patient showed relapse by progredient pleural effusion (Figure 3). Pleural effusion material and a cryofixed liver biopsy were provided for further molecular testing. In the cryofixed tissue sample, HC-NGS confirmed the EGFR and TP53 mutations and the acquired EGFR and MET amplifications but did not detect any additional on- or off-target resistance mechanism. In parallel, from the pleural effusion material, we performed high-sensitivity HC-NGS which was designed for liquid biopsies [20]. The analysis confirmed the EGFR and TP53 and the amplifications in EGFR and MET. In addition, the high-sensitivity analysis revealed a D1246N mutation in MET at the detection limit (0.1%). Since the D1246N mutation confers resistance to crizotinib [11,12], based on preclinical evidence [21,22] and case reports [14,23,24], the patient started to receive a combination therapy of osimertinib and cabozantinib. He responded quickly to the therapy and started feeling much better after a few days. Approximately 8 weeks later, in February 2022, stable disease was shown on CT in the metastatic liver site and partial pulmonary remission, while the general clinical picture and quality of life of the patient were normal (Figure 3).

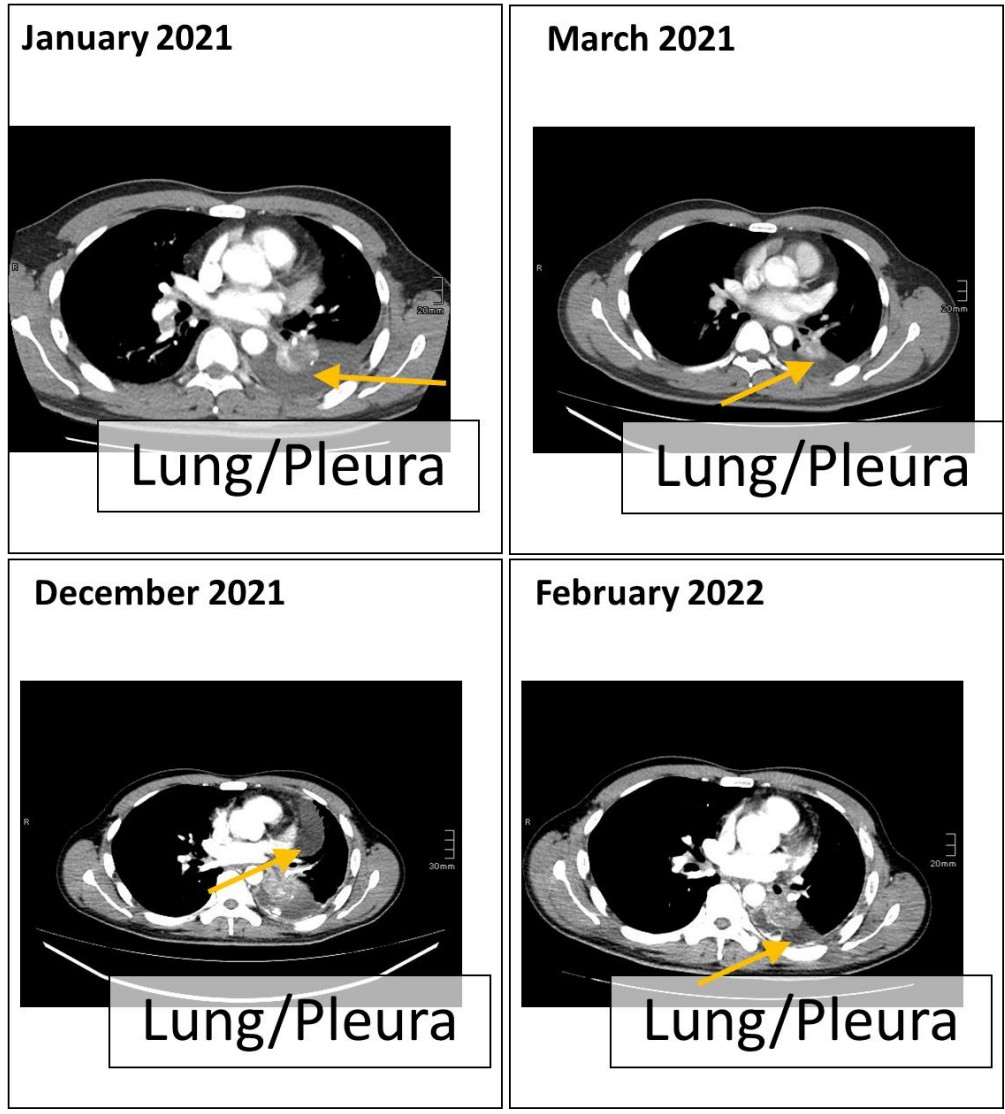

**Figure 3.** Computer tomography of the chest section. The scan was taken from the chest section at the time of tumor progression after osimertinib treatment (January 2021 to March 2021) and after the combined treatment of osimertinib and crizotinib (December 2021 to February 2022).

In May 2022, the hepatic metastases progressed, and another parallel HC-NGS analysis from cryofixed paraffin-embedded liver biopsy material and a liquid biopsy from blood (cf-DNA) were performed. In the cryofixed material (lower left lung, bronchial mucosa biopsies with undermining infiltrates of a minor differentiated adenocarcinoma), the molecular pattern of the 2021 analysis was reproduced without the detection of any point mutation in MET. However, the cf-DNA liquid biopsy HC-NGS confirmed the EGFR, the TP53, and the previously detected acquired D1246N resistance mutation in MET. NGS also detected additional resistance mutations, D1246H and L1213V, in the MET gene. Of note, copy number analysis in this sample in this liquid biopsy assay was not possible.

As no further targeted therapy options were left and due to pulmonary and hepatic progression, the patient started to receive a combination of chemo- and immunotherapy by carboplatin/paclitaxel atezolizumab and bevacizumab. After two cycles in July 2022, we saw further progression in the liver and lung, and treatment was additionally changed to carboplatin and pemetrexed. Unfortunately, the treatment did not succeed. Due to further progression, we decided to stop active tumor treatment in September 2022, and lastly, in October 2022, the patient passed away.

## 3. Discussion

Osimertinib is approved for the first-line treatment of EGFR-mutated NSCLC patients, and molecular alterations causing resistance differ from earlier generation EGFR TKIs [25]. For osimertinib, the most common scenario is the appearance of the EGFR C797X mutation, usually leaving no option for further targeted treatment [26]. In about 12–29% of cases, off-target resistance mechanisms, such as MET amplification, allow rationale for combination treatments [2,3].

In our case report, we present an osimertinib-treated EGFR-mutant lung adenocarcinoma patient with an acquired high-level, focal MET amplification and further point mutations in the MET gene that allowed the combined treatment targeting the MET- and the EGFR-gene.

### 3.1. Applied Molecular Diagnostics

For the molecular diagnostics and follow-up, we used different, complex HC-NGS assays only (Table 1, Figure 1). In addition to the detection of clinically relevant point mutations and gene fusions, some of the applied assays included modules of TMB, MSI, and copy number variation (CNV) analysis [18,20]. When CNV analysis was possible, the assay allowed for focality determination; as non-focal amplifications are often the result of chromosomal aneuploidy and affect a large number of genes, these might not be of diagnostic relevance.

### 3.2. MET Amplification as a Resistance Driver

The high-level focal amplification in MET is described in the context of EGFR-TKI treatment and can probably be considered as the predominant resistance driver to osimertinib in this case. Previous reports of crizotinib and capmatinib response suggest that the amplitude of MET correlates with its function as an oncogenic driver [27,28]. In addition to MET, the EGFR gene was also amplified in the patient, and this potentially contributes to resistance to osimertinib [3]. However, the increased copy number in EGFR seems not to be a focal amplification, and considering gene deletions in numerous other genes in this tumor, it could have been derived as a result of chromothripsis.

### 3.3. TP53 and Genomic Instability and TMB

For a never-smoker with an activating EGFR mutation, TMB was relatively high (17.42 Muts/mB), providing a rationale for immune oncological treatment in the second line [29]. However, since MET amplification appeared to be the main targetable driver in resistance development, instead of second-line checkpoint inhibition, the combination of osimertinib with crizotinib was preferred. This led to a pronounced clinical response

after 6 weeks of therapy and was well tolerated by the patient. Despite the elevated TMB and genomic instability represented in a low degree of MSI, numerous gene deletions, and amplifications, the tumor lacked deleterious point mutations in the genes that have been previously correlated with such alterations in the literature (i.e., BRCA1/2, POLE, ARID1A, or mismatch genes). In addition to the EGFR driver mutation, the tumor only included a deleterious TP53 mutation.

Alterations in TP53 have been correlated in multiple studies not only with poor progression [30] but also with copy number aberrations [31] and an elevated number of point mutations [32–34]. As the p53 protein, considered a genome guardian, plays a pivotal role in controlling genome integrity [35], the probable loss of the functional p53 protein, as a result of the splice mutation and the gene deletion in TP53, might have contributed to the general genomic instability. In this regard, the case also highlights the clinical utility of the regular screening of TP53 mutational status, preferably with NGS.

### 3.4. Tumor Heterogeneity and Hybrid Capture

During the patient's treatment, we performed four analyses on tissue samples (fresh frozen or FFPE), one analysis on pleural effusion material, and one analysis on cfDNA from liquid biopsy. In all of these analyses, both the EGFR exon 19 deletion and the TP53 mutation were detectable. Similarly, the focal MET amplification was also detectable in all samples after osimertinib resistance developed. The only exception was the analysis in May 2022, as CNV was not included in the applied liquid biopsy assay.

Contradictorily, major differences were observed in the pattern of the MET resistance mutations. Point mutations, which might have contributed to the decreased efficacy of MET-TKI therapy (D1246N, D1246H, and L1213V), were not detected in the fresh frozen or FFPE tissue samples, only in cfDNA and pleural effusion. This discrepancy can be explained by the nature of analyses from liquid biopsy and pleural effusion material, as it represents tumor heterogeneity and clonal diversification [2,36,37]. This also suggests that these point mutations developed in other, non-hepatic metastatic sites where no tissue biopsies were taken. In addition, the aggressive development and progression might have resulted from a different resistance mechanism that was not covered by our assays.

### 3.5. Possible Remaining Treatment Options after Cabozantinib

Although the stepwise and combined treatment of EGFR- and MET-TKIs resulted in an overall survival of 31 months for the patient, his rapid progression and latterly developed resistance to osimertinib/cabozantinib treatment raises the question of further therapeutic options that have not been yet established in the clinical routine. Currently, the IN-SIGHT 2 study prospectively evaluates the addition of tepotinib to osimertinib in patients with progressive EGFR-mutant Met amp NSCLC [38]. According to the latest results from the CRYSALIS-2 study, there are encouraging results for the EGFR-MET bispecific antibody therapy of amivantamab and lazertinib [39]. In addition, several preclinical studies suggest simultaneous treatment with type I and type II MET-TKIs, as it may be a clinically viable approach to delay the emergence of on-target MET-mediated drug-resistance mutations [40]. And last, comparative studies might be necessary for the optimal treatment of EGFR- and MET-TKI-resistant lung cancers (i.e., amivantamab/lazertinib vs. osimertinib/cabozantinib).

### 3.6. Patient Survival Benefits and Economic Considerations of Advanced Molecular Diagnostics

Our patient underwent routine molecular testing, including both tissue and liquid biopsies, at multiple time points during their disease progression. These molecular assays are part of our standard portfolio and are available to all patients. We did not employ any additional research-based analyses or modules. While this approach is common in our clinical practice, it is important to note that the cost of liquid biopsy-based disease monitoring is typically not covered by German healthcare providers. However, certain health insurance companies, as part of the German network known as the '*National Network*

*of Genomic Medicine, nNGM'*, do provide reimbursement for NGS liquid biopsies when managing cases of NSCLC where a change in therapy is required due to disease progression. In addition, a recent study has demonstrated that NGS-based analysis offers a cost-effective strategy for advanced NSCLC [41].

## 4. Conclusions

Resistance mechanisms to third-generation EGFR TKI osimertinib often include therapeutically targetable mechanisms, including MET amplification. Due to the complexity of the possible resistance mechanisms to osimertinib, it is crucial to apply complex diagnostic methods, such as HC-NGS, which enable one-step evaluation of the multiple genomic alterations of the tumor, including proper CNV analysis. The combination of osimertinib/crizotinib and osimertinib/cabozantinib provides the next therapeutic opportunity in NSCLC patients with EGFR oncogene addiction following resistance development caused by MET amplification.

**Author Contributions:** Conceptualization, B.J., P.H. and L.C.H.; methodology, B.J., K.A. and J.P.; software, E.M.W.; validation, C.V., J.P. and B.J.; formal analysis, C.V., B.J. and J.P.; investigation, P.H. and L.C.H.; resources, M.T. and P.H.; data curation, J.P. and C.V.; writing—original draft preparation, B.J.; writing—review and editing, M.F., P.H. and L.C.H.; visualization, B.J., P.H. and L.C.H.; supervision, L.C.H.; project administration, K.A.; funding acquisition, M.T. All authors have read and agreed to the published version of the manuscript.

**Funding:** This research received no external funding.

**Institutional Review Board Statement:** The study was conducted according to the guidelines of the Declaration of Helsinki and approved by the Ethics Committee of the University of Oldenburg, No. 2021-126.

**Informed Consent Statement:** Written informed consent was obtained from the patient to publish this paper.

**Data Availability Statement:** Data sharing is not applicable to this article.

**Acknowledgments:** This paper is dedicated to the patient's memory and his willingness to contribute to a better understanding of the complexity of resistance development in lung cancer. The authors would also like to express their appreciation to Stefanie Schatz.

**Conflicts of Interest:** M.T. has received honoraria for consulting and/or lectures from Astra Zeneca, Boehringer Ingelheim, BMS, MSD, Novartis, Lilly Oncology, Roche, and Takeda. M.F. has received honoraria for consulting and/or lectures from Astra Zeneca, Boehringer Ingelheim, Roche, and Novartis. L.C.H. has an advisory role at Agilent, Roche, Astra Zeneca, Lilly, Bayer, and Smart in Media. All other authors declare no conflict of interest.

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
