# Peer review of "The Combined Therapy of Cabozantinib, Crizotinib, and Osimertinib in a Lung Cancer Patient with Acquired MET Amplification and Resistance Mutations"

_curroncol, doi:10.3390/curroncol30100635_

Round 1

Reviewer 1 Report

In the article, the authors presented a report on a patient with EGFR-mutant lung adenocarcinoma and gradually acquired resistance as a result of focal MET amplification, as well as approaches to overcoming tumor resistance. The title of the article fully reflects the content of the article.

In the "Abstract" section, the authors briefly presented the relevance of the study: the search for approaches to molecular diagnostic analyses and personalized treatment of patients with lung cancer with acquired resistance. The section briefly describes a 50-year-old male patient with EGFR-mutant lung adenocarcinoma and gradually acquired resistance as a result of focal MET amplification, as well as the results of combined and sequential treatment. In conclusion, the authors pointed out the importance of well-thought-out sequential molecular diagnostic analyses and personalized treatment of patients with acquired resistance. To understand the essence of the section, it is necessary to indicate that the point mutations of MET D1246H and L1213V were evaluated using next-generation sequencing with hybrid capture (HC-NGS).

The "Keywords" presented in the article correspond to the content of the article and are necessary.

In the section "1. Introduction", the authors examined the antitumor effects of osimertinib, as well as the mechanisms of resistance during administration of the drug. The authors pointed out possible approaches to combating resistance when taking osimertinib. To do this, it is possible to use crizotinib, which also leads to point mutations in the MET gena. Overcoming crizotinib-induced resistance is possible by prescribing cabozantinib. The purpose of this study was to present data on a sequentially treated EGFR-positive patient with NSCLC with gradually acquired resistance mechanisms. Combined and sequential treatment with osimertinib, osimertinib/crizotinib and the third line of osimertinib/cabozantinib was performed.

In the section "2. Case Presentation", the authors presented the patient's medical history and the results of molecular studies using NGS. In addition, the results of treatment with osimertinib, combined treatment with osimertinib and krizotinib (including side effects), osimertinib and cabozantinib, combination of chemo- and immunotherapy with carboplatin/ paclitaxel, atezolizumab and bevacizumab (when changing carboplatin to pemetrexed after two cycles) were demonstrated. Each time, the reason for the change of treatment was the detected acquired mutations of resistance to treatment. Unfortunately, the treatment was not successful, the patient died.

In the section "3. Discussion" the authors analyzed their results, for this purpose published data from other research groups were involved.

In the section "4. Conclusions", the authors pointed out that it is extremely important to use complex diagnostic methods, such as HC-NGS, which make it possible to evaluate multiple genomic changes of a tumor in one step, including proper CNV analysis. The results of such diagnostics make it possible to choose the optimal treatment approach for patients with NSCLC with dependence on the EGFR oncogene after the development of resistance caused by increased MET.

In the section "4. Conclusions", the authors pointed out that it is extremely important to use complex diagnostic methods, such as HC-NGS, which make it possible to evaluate multiple genomic changes of a tumor in one step, including proper CNV analysis. The results of such diagnostics make it possible to choose the optimal treatment approach for patients with NSCLC with dependence on the EGFR oncogene after the development of resistance caused by increased MET.

The article is interesting, timely and important for the clinic. The text of the article is written clearly. The manuscript did not cause any ethical problems. All links to publications in the "References" section are necessary and correct, made in the right style. Of the 40 links that are presented in the article, 31 links over the past 5 years. I have no concerns about the similarity of this article with other articles published by the same authors.

Competing interests of authors do not create bias in the presentation of results and conclusions.

Author Response

Dear Reviewer, we thank you so much for your comments and structured review of our manuscript.  Kind regards, Balazs Jori.

Reviewer 2 Report

A scientific group, including practicing oncologists, described an interesting case that clearly demonstrates the importance and necessity of introducing a genetic apparatus for the dynamic selection of treatment regimens for oncological diseases. Despite the fact that the patient died, nevertheless, the authors of the article managed to extend his life by almost 3 years, which is a fantastic achievement given this form and stage of cancer. I believe that the article should be published as soon as possible. To the authors, as the quintessence of the study, I propose to supplement the article with some kind of flowchart of the general approach to choosing a treatment regimen, which includes genetic phenotyping of the tumor.

Author Response

Dear reviewer, thank you so much for your valid comments. Indeed, the biomarkers that were considered for the therapeutic decision needs to be highlighted for matter of clarity. Therefore, we change the flowchart (Figure 1) by underlining the important biomarkers that we used for the stratification. Kind regards, Balazs Jori.

Reviewer 3 Report

This is a well-described case of unresectable lung cancer (NSCLC stage IVB) in a patient that is finally treated with a combination of TKIs.
The authors follow-up the patient's response to treatment month-by-month and re-examine him using molecular techniques in order to change their targeted treatment.
Only a couple of comments to be addressed:
- This is a wonderful approach to a single patient. But, is this standard of care in your facility? If not, please state that additional techniques were applied for investigational purposes.
- Please add a comment about cost of advanced molecular techniques in relation to benefit (in terms of survival and disease free survival) in the discussion part.

Author Response

Dear reviewer, thank you sou much for your comments and valid points. Unfortunately, German health care providers do generally not cover the sequential use of liquid biopsies as a tool for therapy monitoring. However, some select health insurance companies, as part of the German network “national Network of Genomic Medicine, nNGM”, do indeed reimburse NGS-based liquid biopsies, in case disease progression of NSCLC under targeted therapy requires a change of therapy. We added an explanation to the discussion as you suggested. We also cited a publication showing cost effectiveness of NGS-testing in a recent US-study and benefit for the patient. We also would like to highlight the fact that we only used assays that are available for our routine diagnostics portfolio, and therefore available for every patient. Kind regards, Balazs Jori